# Enhancing Safety on Construction Sites: A UWB-Based Proximity Warning System Ensuring GDPR Compliance to Prevent Collision Hazards

**DOI:** 10.3390/s23249770

**Published:** 2023-12-12

**Authors:** Silvia Mastrolembo Ventura, Paolo Bellagente, Stefano Rinaldi, Alessandra Flammini, Angelo L. C. Ciribini

**Affiliations:** 1Department of Civil, Architectural, Environmental Engineering and Mathematics (DICATAM), University of Brescia, 25123 Brescia, Italy; silvia.mastrolemboventura@unibs.it (S.M.V.); angelo.ciribini@unibs.it (A.L.C.C.); 2Department of Information Engineering (DII), University of Brescia, 25123 Brescia, Italy; paolo.bellagente@unibs.it (P.B.); stefano.rinaldi@unibs.it (S.R.)

**Keywords:** construction safety, proximity warning system, collision accidents, sensing technologies, ultra-wideband, data privacy, data protection, general data protection regulation

## Abstract

Construction is known as one of the most dangerous industries in terms of worker safety. Collisions due the excessive proximity of workers to moving construction vehicles are one of the leading causes of fatal and non-fatal accidents on construction sites internationally. Proximity warning systems (PWS) have been proposed in the literature as a solution to detect the risk for collision and to alert workers and equipment operators in time to prevent collisions. Although the role of sensing technologies for situational awareness has been recognised in previous studies, several factors still need to be considered. This paper describes the design of a prototype sensor-based PWS, aimed mainly at small and medium-sized construction companies, to collect real-time data directly from construction sites and to warn workers of a potential risk of collision accidents. It considers, in an integrated manner, factors such as cost of deployment, the actual nature of a construction site as an operating environment and data protection. A low-cost, ultra-wideband (UWB)-based proximity detection system has been developed that can operate with or without fixed anchors. In addition, the PWS is compliant with the General Data Protection Regulation (GDPR) of the European Union. A privacy-by-design approach has been adopted and privacy mechanisms have been used for data protection. Future work could evaluate the PWS in real operational conditions and incorporate additional factors for its further development, such as studies on the timely interpretation of data.

## 1. Introduction

Safety on construction sites is an issue of constant topicality, as the construction industry is notoriously one of the most dangerous for worker safety [1], as also confirmed by international statistics. Previous studies have estimated that at least 60,000 construction workers die each year [2], and in industrialised countries, the construction industry accounts for 25–40 percent of all workplace fatalities, despite representing only 6–10 percent of the global workforce [3]. According to statistics from the U.S. Bureau of Labor Statistics [4], in 2021, 986 workers in the private construction industry suffered fatal work injuries in the United States, accounting for 19 percent of all workplace fatalities. Similar figures can be found in Europe where, according to statistics published by Eurostat [5], more than a fifth (i.e., 22.5 percent) of all fatal accidents at work occurred in the construction sector, which ranks first among the productive sectors causing fatal accidents among the European working population. Observing the issue on a national scale, according to INAIL (Italian National Institute for Insurance against Accidents at Work) [6], the construction sector is one of the most closely monitored due to the high risk associated with activities that involve a considerable physical effort (e.g., carrying heavy loads, working in uncomfortable positions and standing for long periods of time), carried out in uncomfortable environments and sometimes under adverse climatic conditions (e.g., summer’s high temperatures that subject workers to great stress). Data about the Italian context show how the sector ranks first for fatal injuries, with 196 fatalities in 2021, confirming the hazardous nature of construction [6].

Despite the importance given to construction safety management, traditional method, and preventive measures—e.g., risk analysis, worker training, site inspections and quality control checklists—do not always ensure a significant improvement in safety levels [7,8]. Workers tend not to perceive and recognise many of the hazards on construction sites as potential sources of risk, even when all of the above measures are effectively implemented [8,9,10]. Maintaining the appropriate level of situation awareness, which is critical to safety, is therefore a challenge [11]. Within the context just described, collision accidents involving operators and moving construction vehicles are one of the main safety issues in construction workplaces, being one of the leading causes of non-fatal accidents and one of the most frequent causes of death on construction sites internationally [12,13,14,15,16]. The dynamic characteristics of construction site activities may be the cause of collision accidents, with construction resources often moving in situations of excessive proximity [8]. Inadequate analysis of spatial interferences in the design of construction site layouts often results in construction site spaces that are, at least temporarily, inadequate for the safe conduct of operations [17,18]. In addition, other factors can be identified that increase the likelihood of workers becoming victims of collision accidents: for example, the possible loss of concentration in performing daily activities due to fatigue and repetitive tasks, and the reduced visibility of construction vehicle operators due to blind spots [8,12,13,14,15,16]. Statistics, in fact, often highlight how collision fatalities can be caused by the failure of the operator to check the position of workers in the manoeuvring area, leaving workers undetected in blind spots or too close to moving vehicles [15]. In addition, inadequate safety equipment on construction vehicles (e.g., rear-view mirrors and reversing horns) and inadequate signposting of pedestrian routes in construction site layouts have been also identified as risk factors for collision accidents [14,15].

In order to support the increasing attention to the role of collision accidents for the safety of construction workers, the use of sensing technologies and tracking systems is discussed in the literature as a way to promote continuous safety monitoring for early risk detection and timely prevention of collision accidents. In this sense, previous studies have discussed the possibility of providing workers with real-time alerts and warnings of potential hazards. The development and use of proximity warning systems (PWSs) has been investigated as a way of identifying hazardous situations on a construction site, improving worker safety and helping to prevent construction injuries, e.g., [12,13,14,19].

The main contributions of this paper with the respect to the state of the art are as follows: Analyse the privacy issue raised by a solution used to monitor workers for safety reasons in construction sites;Analyse the practical use cases of security in construction sites with particular focus on PWS;Perform an experimental evaluation of the risk conditions in the interaction between workers and construction machineries in construction sites;Propose a privacy-by-design solution able to satisfy the requirements of GDPR;Apply the proposed PWS approach to the identified use cases.

This paper describes the design of a sensor-based PWS, primarily aimed at small and medium-sized construction companies, for the real-time detection of potential risks to construction workers due to excessive proximity to a moving construction vehicle. It takes into account several factors mentioned in the previous literature, such as implementation cost, the actual nature of a construction site as an operating environment and data protection [9]. The assessment of the proposed technology for an accurate estimation of the proximity is out of the scope of the current paper. More details about the assessment of the PWS technology are provided in [20].

The rest of this paper is structured as follows. Section 2 describes the research background of this study; Section 3 illustrates the application case, the methods and materials applied, while Section 4 presents the PWS design and preliminary implementations, and Section 5 concludes the paper with concluding remarks, limitations of this study and future developments.

## 2. Research Background

### 2.1. PWS: Available Technological Solutions

PWSs use real-time positioning systems to identify the location of workers in relation to potential collision hazards (e.g., proximity to moving construction vehicles). In addition to risk detection, PWSs alert workers by sending notification signals when their safety is at risk due to the excessive proximity to a dangerous situation (i.e., to prevent the accident from occurring). This results in improved situational awareness of the workers, who should then be able to take appropriate action to avoid risks once they have received the warning signal [11]. 

Several research papers have examined technological advancements to monitor the safety of workers on construction sites and demonstrated the applicability of sensing technologies for the real-time detection of proximity-related safety hazards. In more detail, real-time locating and tracking technologies are becoming widely used for the continuous monitoring of construction resources (workers, materials and equipment) on site [11]. The aim is to prevent the exposure of workers to hazards and potential accidents, in addition to increasing their situation awareness [11,21], by detecting and alerting construction workers and vehicle operators in real time [22,23]. In addition, using a PWS to track the location of construction site resources in real time also allows the understanding of how long workers are in a specific location, information which can then be used to assess how much risk an individual is exposed to [24]. 

Sensor-based locating technologies (e.g., Global Positioning System (GPS) [25], Radio Frequency Identification (RFID) [26,27,28,29] and ultra-wideband (UWB) [30,31]) and vision-based sensing technologies [32,33,34] have been considered as effective methods for the advancement of construction safety management in this sense [35]. According to Soltanmohammadlou et al. [22], who studied applications of Real-Time Locating System (RTLS) technologies in construction safety management, RFID, vision-based locating technologies and UWB are the most widely used for safety purposes in construction. Radar-based, WiFi-based and infrared technologies appear to be the least implemented ones in the literature, while examples of the application of GPS and Bluetooth low energy (BLE) RTLS are also available [21,36] (Table 1).

### 2.2. Introduction to UWB Technology

Narrowing down the scope of the application of sensing technologies to the detection and prevention of collision accidents, the precise positioning capability of different locating sensor-based technologies is a key factor to consider [21]. Previous studies demonstrated how the adoption of UWB technologies to collect spatial–temporal data about the movement of construction resources (workers, materials and equipment) could address this need [37,38,39,40]. 

UWB technology is typically used for automatic asset monitoring in indoor environments (e.g., logistics), but it can also achieve a range of 100 metres in outdoor settings. Additionally, it does not require extra hardware, like routers, to run. The UWB communication protocol uses radio wavers for short-range (up to two hundred metres), high-bandwidth (at least 500 MHz) and high-data-rate communication, and it is intended to transmit signals with spectrally dispersive modulation over extremely large bands (i.e., 500 MHz or more) [41]. Figure 1 depicts a typical UWB RTLS network architecture. Network nodes can be referred to as anchors, set up at fixed places, and tags, fastened to the moving objects that need to be tracked [8]. The node in charge of managing transmission timing between nodes and permitting new nodes to join the network is known as the network coordinator. Anchors are normally installed at a maximum distance of 25 metres, and they broadcast radio pulses to initiate the positioning process. Depending on the technology available, positioning typically has an error of no more than thirty centimetres and an update rate of every fifty milliseconds. Unless solutions like GPS receivers with Real Time Kinematic (RTK) support are used, it is difficult to obtain these results using GPS technology. Moreover, the cost of these solutions would make them unsuitable for widespread use on construction sites [8]. On the other hand, studies have shown how the performance of UWB-based PWSs should be evaluated, taking into account aspects such as the error rate of real-time location tracking and the possible effects of non-line of sight [40,41]. Limitations can also occur if the environment where the monitoring takes places goes through significant changes that, for example, prohibit signals from being received. In addition, this method necessitates manual intervention for anchor positioning and RTLS system registration for tag trilateration [8,37,38].

UWB technologies have been widely used for this purpose in the construction sector, allowing the location of workers and the tracking of construction resources (e.g., equipment and materials) in real-time. This possibility, in fact, supports the continuous monitoring of the distance between a worker and an approaching construction vehicle [37,38]. UWB technologies have been adopted for real-time location tracking of resources in cluttered [39] and harsh [40] construction environments. Studies have also assessed the actual reactivity of workers in taking action to protect their own safety once they have received an alert [25]. The question has been asked, for example, as to whether the risk of an overabundance of audible alerts might actually be counterproductive to the safety of workers who, with an overabundance of information, would find themselves ignoring them. To this end, Chan et al. [19] incorporated worker awareness in the development of a PWS in order to reduce redundant alerts by means of GPS, UWB and Inertial Measurement Unit (IMU) sensors attached to a hard hat to collect the orientation of workers in addition to the real-time location to generate an alert only if the worker is not looking at hazards within a hazardous distance. Another aspect to consider is how the adoption of such technology integrates with the dynamism typical of construction progresses. Existing solutions usually require a fixed UWB infrastructure [37,38,39,40]. Pittokopiti and Grammenos [13], instead, discussed the implementation a UWB collision avoidance system that does not require a fixed infrastructure precisely in order to take into account the need to frequently reconfigure the layout of a construction site according to the work schedule. The proposed solution adopted a low-cost approach and a battery-powered system that was able to detect in real time the risk of collision accidents if combined with a linear regression algorithm. Their results demonstrated a robust UWB communication link under line-of-sight conditions and in indoor environments. In these conditions, the system was able to detect hazards up to 91 m and to guarantee workers the needed time to take appropriate actions for their safety. However, in this case, a remote-controlled toy car was used to assess the collision avoidance algorithm instead of real construction vehicles.

### 2.3. PWS: The Open Issues

Although previous studies have demonstrated sufficient accuracy for the development and implementation of sensor-based PWSs, a number of issues remain to be addressed [9]. First, the dynamic nature of construction sites needs to be considered. It requires the continuous reconfiguration of construction areas according to the construction schedule and the activities that have to take place. A sensor-based PWS must be able to respond to changing construction site conditions as construction progresses by providing adequate redundancy or the ability to reconfigure the system if necessary [9,13]. When UWB localisation and communication systems are adopted, there is the need to consider that they have to completely surround the monitored construction site area, always taking into account that they might be sensitive to occlusion and blind spots. In addition, sensor-based PWS deployment, which in the literature is primarily evaluated in controlled environments, must take into account the need to properly mount and encase tags and anchors to protect them from exposure to dust, rain and elevated temperatures in actual construction sites. Moreover, the timely interpretation of data gained from sensors on construction sites in real time is a major concern to support decision making. 

Finally, concerns about worker privacy and data protection have recently emerged as one of the issues hindering the use of sensing technologies and, specifically, wearables in the construction sector [42,43,44]. The literature reports how data privacy is sometimes ignored in favour of highlighting the potential to improve construction safety and other aspects, such as productivity and efficiency, through the use of cutting-edge smart technologies and digital breakthroughs. However, previous studies investigating users’ intentions to adopt wearable devices have highlighted that the lack of guaranteed privacy of workers’ personal data has been identified as a significant limitation of modern technologies and one of the reasons why users are reluctant to adopt them [43,44], along with perceived utility [45]. Workers resent the concept of being constantly monitored in the workplace, and they may feel insecure when the data collected, including personal data such as their name and real-time location, both during working hours and rest time [42,45], could potentially compromise their privacy and be used by employers to evaluate their performance in terms of productivity [45]. 

As wearable devices collect and wirelessly transmit personal data to a receiver (e.g., a smartphone), data security and encryption have emerged as key challenges in the development of wearable technology [44]. In this sense, Xu et al. [42] highlight that in order to facilitate the adoption of sensing technologies on construction sites, particularly wearable sensing technologies, it is necessary to separate the management of non-sensitive safety performance data from privacy data. Moreover, the research has suggested that workers’ willingness to accept tracking technologies may be supported by a deeper understanding of the real benefits that workers derive from the collection of these data. Among them is the fact that these data may serve as a reliable resource for the instantaneous identification of risks associated with a particular worker or work setting [44]. In such a context, Rao et al. [9] recommend that research and studies be conducted to determine the best way to incorporate privacy protections into monitoring systems [9].

The main indication about how to implement such protection comes directly from Article 25 of EU GDPR [46] as “the controller shall, both at the time of the determination of the means for processing and at the time of the processing itself, implement appropriate technical and organisational measures, such as pseudonymisation, which are designed to implement data-protection principles, such as data minimisation, in an effective manner and to integrate the necessary safeguards into the processing in order to meet the requirements of this Regulation and protect the rights of data subjects”. At the same time, “the controller shall implement appropriate technical and organisational measures for ensuring that, by default, only personal data which are necessary for each specific purpose of the processing are processed”. Those principles, generally known under the names of privacy-by-design and privacy-by-default, were already investigated in many application fields such as electronic identification (eID) [47], electronic health records [48], and Smart Buildings [49]. Nevertheless, to the best of authors knowledge, no one has disclosed privacy-by-design insights in applying PWSs to construction site monitoring and management.

Thus, the aim of the research presented in this paper is to analyse and to provide a response to two of the open issues in adopting the PWS in construction sites: the dynamic nature of the construction site (and thus the capability of the PWS to quickly reconfigure itself in accordance with the activities anticipated in the construction schedule), and the fundamental role of data privacy and data protection (under the rule of GDPR normative) as an integral part of the implementation of digital and data-driven processes. Particularly, this research does not focus on the use of technologies or frameworks to assess GDPR compliance as in [50] but rather on the design process required to identify the set of information needed to operate PWSs in construction sites and the system requirements to implement a proper privacy-preserving system compliant with the EU GDPR.

Furthermore, the need to keep the entire cost of adopting the PWS low in order to encourage its adoption by construction Small Medium Enterprises (SMEs) has been considered.

## 3. Research Methods and Materials

As mentioned in the previous section, UWB technology is the most promising solution currently available for the design of a PWS because it provides high precision location tracking; low power consumption; high data rates; immunity to interference; low latency; scalability; low cost. However, a full adoption of UWB technology for PWS design still requires further investigations, whose target is to take into account the specific characteristics of construction site environments. 

Research activities were organised in six macro-phases, with some of them overlapping during the testing in order to proceed with an iterative process of data collection and analysis (Figure 2).

Phase 1—Analysis of the use cases regarding the adoption of PWS in construction site for safety management;Phase 2—On-site testing for real-time proximity analysis;Phase 3—Design of the system architecture;Phase 4—Iterative validation of the system architecture;Phase 5—Verification of the compliance with GPDR.

An empirical study was developed involving the direct participation of potential end-users (e.g., earthmoving machine operators) of the proximity working system, with the aim of both analysing a potential application case (i.e., the use of a front loader in typical construction site activities involving the interaction between the construction vehicle and workers) and obtaining feedback on the proposed technological solution. Social partners and representatives of management and labour have been involved in regular update meetings on the development of the research in order to share with them updates on the selected technologies and the related possibility of collecting, processing and storing workers’ position data. Their perspective on the collection of workforce data, which is necessary to track their position on-site in real time as a necessary condition for the operation of a PWS, was necessary firstly to understand the hesitations about the adoption of location tracking tools and secondly to validate the proposed solutions with a view to their adoption by workers. 

The tests were not conducted on a real construction site but in a controlled environment using real construction equipment and involving potential operators. In the controlled environment, an actual construction site area was simulated, albeit under simplified conditions in terms of the number of moving construction vehicles, operators, noise, etc. Construction activities that typically involve the proximity of earthmoving construction vehicles and workers (e.g., excavation and backfilling of material and ground levelling) were considered to formulate preliminary assumptions on how the PWS was supposed to work. Two distinct types of earthmoving construction vehicles were considered for the application case. They were selected to take into account their different characteristics in terms of geometry and interaction with workers (Figure 3): (1) a front-end loader (i.e., Takeuchi TL6R) as an example of a vehicle with a fixed geometry, and (2) an excavator (i.e., Takeuchi TB640) as an example of a vehicle with variable geometry (i.e., the variable configuration during movement, with the arm retracted and the arm extended) [8]. Technical specifications of the selected construction vehicles and the direct experience of the involved end-users formed the basis on which we devised the initial hypothesis of acceptable vehicle–worker distances to set the PWS’s operating rules in terms of ensuring safety for workers but at the same time, the productivity of the construction site (Figure 4). As described in Mastrolembo Ventura et al. [8], five areas have been identified as a first rough guideline for setting the PWS:Black area, (i.e., the operating radius of the construction vehicle). Within this area, there is an immediate risk of serious or fatal injury.Red area, (i.e., an area of two metres around the operating radius of the construction vehicles. According to the manufacturer’s instructions, no person may be present within the red area when the construction vehicle is in motion). In this area, immediate action is required to avoid permanent injury or fatal accidents.Yellow area. Work can be conducted in this area, but caution is advised as the risk can quickly change to red and black.Green area. Within this area, workers should be notified about the presence of moving construction vehicles, but they can work safely.White area. In this area, risks of collision accidents are not detected.

Having contextualised the application case, the design of the PWS started. One element that was considered from the outset was the potential users of this technology within the scope of this research: small and medium-sized construction companies. Therefore, the cost factor guided the initial implementation decisions. To meet this need, a real-time positioning system was first proposed using commercially available modules. This choice allowed an optimal trade-off between location accuracy and equipment cost, making it a suitable solution for accurate distance measurement on construction sites. The reasonable cost of the equipment made it possible to widen the pool of potential construction companies that could adopt such a system, including small and medium-sized ones. The system was scaled up to support multiple anchors and tags in order to ensure a robust UWB communication link also under non-line-of-sight (NLOS) conditions.

A second factor considered was the concept of the construction site as a dynamic operational environment. For this reason, and in order to develop a PWS capable of supporting the reconfiguration of the construction site based on the progress of planned construction activities, two configurations of the PWS system were developed. The first configuration developed envisaged the use of a fixed infrastructure (i.e., stationary UWB anchors in addition to equipment carried by workers and machines). Subsequently, a second configuration of the PWS was developed that allows the implementation of an infrastructure-less system (i.e., without fixed anchors). The infrastructure-less approach is due to the need to overcome the limitations of using such technologies in a dynamic operating environment, such as a construction site that changes its layout several times during the construction process. 

Finally, a third factor was crucial in guiding the design and implementation of the PWS: the adoption of data protection mechanisms. In fact, the ongoing discussion with the social partners revealed that this issue—even more than the technological one—was a discriminatory factor in the adoption or non-adoption of a worker location service such as the one of a PWS. A privacy-by-design approach was adopted to ensure the system’s compliance with the European GDPR, while data protection is generally neglected in similar solutions proposed in the literature.

## 4. System Design and Implementation

The goal of the PWS is to simultaneously detect the position of both construction vehicles and workers in relation to themselves. The system should be then able to calculate the level of risk for collision accidents at each position.

### 4.1. Real-Time Location System

A real-time positioning system has been engineered to achieve an optimal equilibrium between the precision of location tracking and the cost-effectiveness of the devices involved. This system, denoted as the “PWS,” leverages the commercially accessible MDEK1001 Kit [51], from Qorvo—Greensboro, NC, USA, primarily designed for the evaluation of the DWM1001 module and the seamless deployment of a Real-Time Locating System (RTLS). The DWM1001 module is an amalgamation of critical components, encompassing the DWM1000 Ultra-Wideband (UWB) Transceiver, from Qorvo—Greensboro, NC, USA, a nRF52832 microcontroller unit (MCU), from Nordic Semiconductor—Trondheim, Norway, fortified with Bluetooth Low Energy (BLE) capabilities, and a triaxial accelerometer [52]. Notably, the transceiver adheres to the IEEE802.15.4z [53] UWB physical layer standards.

The architecture of the infrastructure realised through the utilisation of the MDEK1001 module is elucidated in Figure 5. It notably incorporates four distinct node classifications: (i) anchors, which constitute static nodes serving as benchmarks for the localisation process; (ii) tags, denoting mobile nodes that necessitate attachment to objects targeted for localisation; (iii) bridges, comprising nodes designated as gateways with the potential to interconnect the RTLS with an Internet Protocol (IP) network; and (iv) external devices, characterising nodes facilitating direct information exchange with the DWM1001 module through a Bluetooth Low Energy (BLE) link. For the establishment of a functional RTLS, it is mandatory that three to four anchors are concurrently within the visibility range of the tag reserved for localisation. Within this network framework, a single anchor is specially configured as the “initiator”, assuming the role of network coordinator for the entire system.

The DWM1001 module inherently accommodates the Positioning and Networking Stack (PANS) firmware, which substantiates the implementation of the anchor node network and orchestrates the bilateral ranging exchanges with the tag nodes. Consequent to this system configuration, each individual tag autonomously computes its precise location through the utilisation of the PANS firmware, which is pre-installed on the MDEK1001 nodes [51].

### 4.2. PWS Infrastructure Architecture: Structured Site

A conceptual framework for a proximity detection system, as depicted in Figure 6, has been realised [20]. The fiscal implications of infrastructure are invariably contingent upon the quantity of its fixed constituents. Consequently, a judicious selection of three anchors has been made. These constraints in anchor placement, along with spatial limitations, have led to the demarcation of a square surveillance area, each side spanning fifteen meters. Within Figure 6, one anchor has been reserved as the network coordinator and is denoted as ‘node C’.

Each worker has been outfitted with a distinct tag, while the allocation of tags to construction machinery has been attached following the geometric attributes of the machinery itself. Specifically, a single tag is used for the front-end loader, while the excavator, characterised by its mutable geometry, necessitates two tags—one affixed to the primary body and the other appended to the extremity of the arm to monitor its deployed or retracted state.

All tags emit UWB signals, which are intercepted by all anchors. These tags establish connectivity with the worker’s personal device, typically a smartphone, through a Bluetooth Low Energy (BLE) link. Tag positions are captured at intervals of one hundred milliseconds by the worker’s device, which subsequently undertakes required calculations to verify proximity between workers and potential machineries. In the event of necessity, the device issues a notification signal to notify the worker. Specifics pertaining to the nature of notification signals have been intentionally omitted within the scope of this paper. The personal device plays a pivotal role in ensuring safety, as it is imperative that emergency notifications are locally computed to circumvent complications that may arise due to communication latency. In addition to its safety functions, the worker’s personal device serves as the communication channel for the RTLS, transmitting data via the MQTT protocol over a WiFi network, thereby facilitating the storage of transmitted location data in a suitable database for subsequent analytics.

The limitation inherent in this system pertains to the reliance on the personal devices of workers to serve as intermediaries for tag connectivity. This limitation may be rectified through the utilisation of the BLE beacon mode, thereby taking advantage of personal devices as a crowd of receivers. Although the feasibility of this approach could not be empirically assessed due to the incapacity of the DWM1001 to emit BLE beacons, it is noteworthy that the radio range and data rate characteristics of the devices employed in the experiments are compatible with the exigencies of BLE beacon implementation, thereby rendering the proof-of-concept theoretically viable [54].

This implementation furnishes a framework for precise object localisation within the confines of a construction site, thereby facilitating safety measures, management oversight and progress evaluation. It is imperative to highlight the susceptibility of UWB technology to electromagnetic reflections and obstructions, which may result in potential blind spots. Meticulous placement and scrupulous management of anchors constitute imperative considerations, particularly in safety-critical scenarios.

### 4.3. PWS Architecture: Infrastructure-Less Site

The actual nature of a construction site as an operating environment is the second factor considered for the development of the PWS. In fact, costs, environment or operational constraints could impede the deployment of anchors and their consequent management. In addition to the infrastructure site configuration, an infrastructure-less configuration (i.e., without stationary anchors) has been developed to eventually overcome the limitation related to the adoption of stationary anchors in a dynamic operating environment such as a construction site. In such cases, it is possible to renounce the absolute localisation of objects on construction sites—as it is possible to do with the infrastructure configuration—and narrow down the scope of data collection to the distance between a worker and surrounding dangers. Figure 7 shows the system architecture in case it is not possible to use fixed anchors on site.

Devices and measurement principles are the same as in Section 4.1 and Section 4.2. In this case, tags have been equipped to the construction machinery while workers have been equipped with anchors. Using the two-ranging method, via the embedded PANS firmware, tags can identify their location with respect to anchors. It is obvious that, as anchors are moving freely in the site and the global reference system is moving with them, such localisation data is useless, but it needs the measure of the distance between any tag and each anchor to be calculated. As tags are attached to the machinery, it constantly knows the distance between itself and any detected worker while working.

### 4.4. Privacy-by-Design Approach for PWS Development

The third factor considered in the development of the PWS is privacy and data protection. Privacy issues have been considered in the context of the need for greater transparency in how employee-related data are collected, stored and used in order to promote the adoption of digital innovation to effectively support worker safety [13,42,43,44,45]. It should be noted that the employer cannot have access to the worker’s personal data, including his or her position on the site, if this could somehow provide additional information about his or her productivity, due to the potential conflict with the worker’s statute.

It is then necessary to adopt pseudonymisation techniques to make it difficult to trace back to the identity of the worker, who, in this use case, should not be supervised in terms of location on the construction site but warned in case of danger to his or her safety. Attention had to be paid to this point, as it was felt that it might not be sufficient to adopt a technique of hashing the worker’s name. This data, in fact, although masked by a seemingly insignificant identifier, could still be matched to the worker based on his or her possible qualifications (e.g., the ability to manage a particular piece of equipment) or other factors. In other words, in designing the PWS, the following question was asked: is it still possible to trace the identity of the worker? If so, it was considered not sufficient to “mask” his or her name but instead to understand based on what data this was identifiable and by whom.

To address this aspect, a privacy-by-design approach was adopted for developing the system architecture of the PWS. The proposed reasoning is valid for both the infrastructure and the infrastructure-less configurations and the related solutions. However, the scenario which poses the most critical concerns about data privacy is the infrastructure case. Indeed, it collects the absolute location of each object monitored—workforce included—with respect to the whole construction site, allowing complete real-time tracking that can be potentially used as a surveillance system. On the contrary, the infrastructure-less solution is only able to indicate the relative position between two entities collecting data only on their mutual interaction, without a single clue on their actions with respect to the construction site and thus to workers’ duties and productivity.

For these reasons, the infrastructure site configuration, which shares information with a central system that manages information at the organisation level and to which data are transferred for later analysis, has been used as a reference case to evaluate issues about data privacy. The first step for a privacy analysis is to identify the type of data processed by the computer system. The following types are considered:

Relative distance: The value defining the relative distance is calculated from the network of anchors positioned on the site. Therefore, data are potentially available to anyone. The information identifying this distance to a type of actor is available to the personal device of the worker but not to the user of the central system. The latter is only notified of the presence of vehicles according to the proximity rules identified.Operator identity: Nobody is allowed to identify the worker using the PWS during normal operations. According to the GDPR in force in the European Union, only the so-called DPO (Data Protection Officer) has the authority to identify and correlate data originating from an actor (i.e., the worker) with his or her actual person. The PWS has been designed by implementing this requirement.Machinery identity: Each construction vehicle is classified according to the risk it introduces to the construction site since the explicit identification of a construction vehicle could lead back to the identity of the worker who is operating it. A vehicle class is defined with the couple conventional classification of the vehicle—number of configurations it can assume (e.g., (excavator—2), (front-end loader—1)).Entity definitions: Each entity with which system actors (in this case, workers and machineries) interact should be defined to be present in the digital representation (i.e., the data model representing each system actor). Details about the interaction could lead to the identification of the subject who interacts and thus lead to privacy violations.Interaction events: each interaction event identified by the system (as dangerous proximities, interdicted area crossing, etc.) could lead to a privacy violation depending on the detail of the information available to whom the event is notified.

The construction vehicle and the worker detect the position of each other. Once the distance rules have been applied to evaluate the risk, the central system receives the relative position. The only difference between a construction vehicle and a worker is the computing power available to the device equipped by the vehicle, which attempts to detect all nearby actors (workers and other construction vehicles). To ensure a design that protects privacy, an assessment has been made for each information looking for potential privacy threats to define suitable specifications. In the particular scenarios investigated in this study, it has been important to identify, for each type of data considered, the following: (a) the origin of the data, (b) how it is manipulated, (c) if, how and where it is stored, (d) the kind of treatments it is subject of, (e) the level of access, (f) information obtained by the data and (g) information not obtained by the data.

The categories of personal data to be protected that are identified by the GDPR as well as the expected allowed operations are reported in Table 2. The expected operations not relevant for the analysed use case have been crossed out.

In Table 3, each data category identified for the considered use case is classified, and the expected operation is marked using the definitions contained in Table 1.

The architecture of the system constrains the storage of user-generated data on personal devices, and only in a second stage does the device attempt to update the remote system that holds the location of all tags assigned to the construction site. In accordance with the GPDR’s privacy requirements, the update imposes an anonymisation phase on the data to be stored. This procedure consists of generating a unique reference each time:The app is installed;An authorised account is logged in;A unique reference is created for each type of data;Location and proximity;Service status;Processing stage.

A dedicated database, with dedicated and different credentials, is also used for each type of information. To guarantee the integrity of the transmission, the HTTPS protocol has been used via oAuth2 authentication. Each user will have specific credentials and access based on a Role-Based Access Control (RBAC) schema.

## 5. Conclusions

This paper describes the design of a prototypal low-cost, ultra-wideband (UWB)-based and GDPR-compliant proximity detection system able to work with and without fixed anchors. The PWS is addressed to small and medium-sized construction companies to collect real-time data from the construction site to monitor the risk of collision accidents involving construction vehicles in motion and the workforce. The design of the PWS has considered, in an integrated manner, factors such as the cost of deployment, the actual nature of a construction site as an operating environment and data privacy. 

Commercially available modules were used. This choice has allowed an optimal trade-off between location accuracy and device cost, making it a suitable solution for accurate distance measurement on construction sites. The reasonable cost of the instrumentation makes it possible to widen the pool of potential construction companies that can adopt such a system, possibly including small and medium-sized construction companies. From a construction management perspective, both an infrastructure (i.e., with fixed anchors) and an infrastructure-less (i.e., without fixed anchors) scenario were analysed. On the one hand, the infrastructure scenario should be integrated with the evaluation of the positioning of UWB fixed anchors in the design of the construction site layout and the schedule of construction activities.

This would allow construction managers to consider the installation of stationary anchors to effectively ensure signal coverage which is always consistent with the current activities. On the other hand, the infrastructure-less scenario has been developed to consider the actual nature of a construction site as an operating environment by eventually overcoming the limitation related to the adoption of stationary anchors. In such cases, instead of detecting the absolute localisation of objects, as done with the infrastructure configuration, it is possible to narrow the scope of data collection to the distance between a worker and surrounding dangers (e.g., moving construction vehicles). Finally, by adopting a privacy-by-design approach, the PWS has been developed in accordance with the GDPR of the European Union. Categories of personal data and expected operations for the use case have been identified and managed, including options for data accessibility by employers, workers or data protection officers.

Future research could assess the PWS under real conditions on construction sites, which are characterised by common disturbances and noise. Moreover, additional factors could be integrated for its further development such as the timely interpretation of data that is required to maintain the user communication loop. This relates to both human–computer interaction and signal recognition (i.e., the ability of the worker, in this case, to understand the signal received by the PWS). In addition, if the PWS detects excessive proximity to a moving construction vehicle, a suitable warning system might be implemented to alert workers so that they can take the necessary mitigation measures. The noise and disturbance conditions typical of construction sites, as well as the possibility of worker distraction in the event of excessive alerts, should be taken into account when developing these warning systems. Furthermore, cooperation with manufacturers could make it possible to directly intervene on construction vehicles in order to optimise the technological capabilities of the PWS.

## Figures and Tables

**Figure 1 sensors-23-09770-f001:**
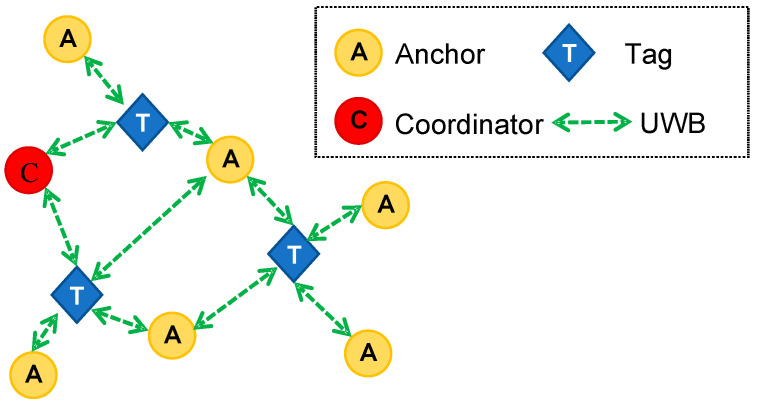
The typical architecture of a UWB RTLS network [8].

**Figure 2 sensors-23-09770-f002:**
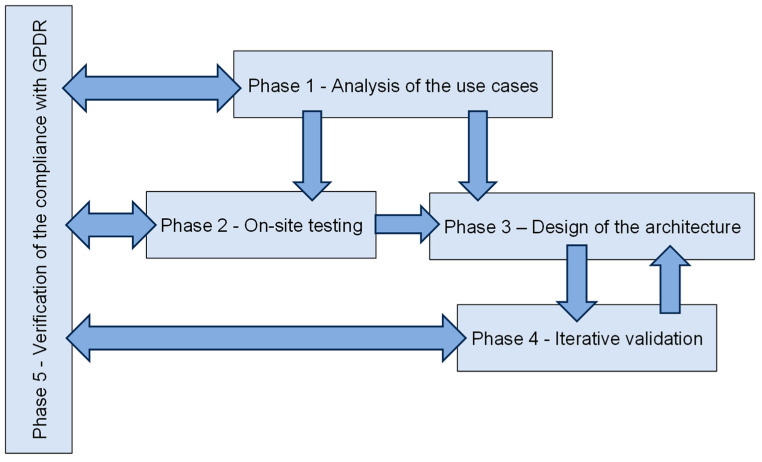
The workflow diagram of research phases.

**Figure 3 sensors-23-09770-f003:**
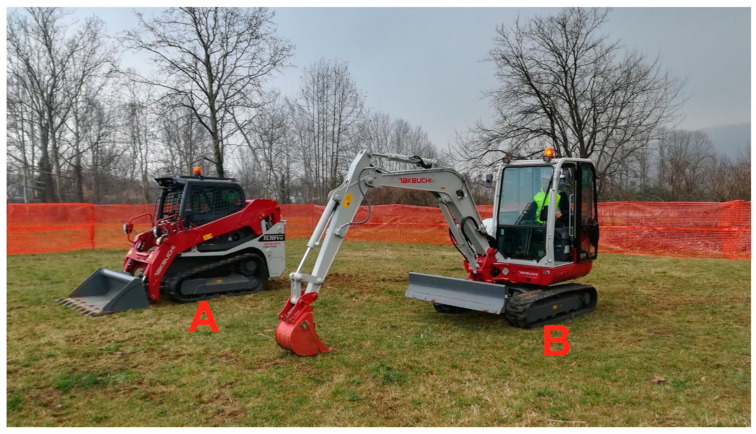
Takeuchi TL6R front-end loader (**A**) and Takeuchi TB640 excavator (**B**) used in the research project to simulate construction activities.

**Figure 4 sensors-23-09770-f004:**
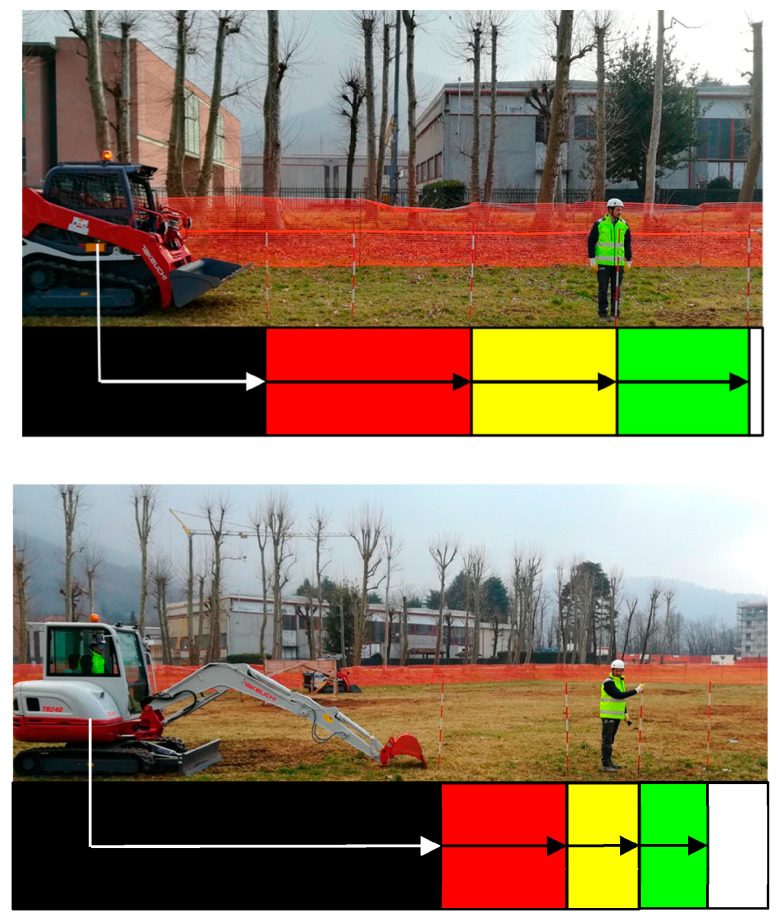
Preliminary assumptions for safety distances between an earth-moving construction vehicle and worker in its proximity [8].

**Figure 5 sensors-23-09770-f005:**
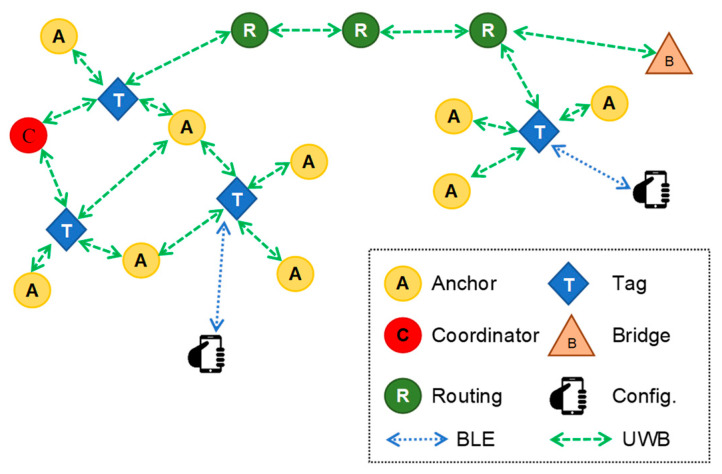
The DWM1001 Real-Time Location System (DRTLS) network and its components.

**Figure 6 sensors-23-09770-f006:**
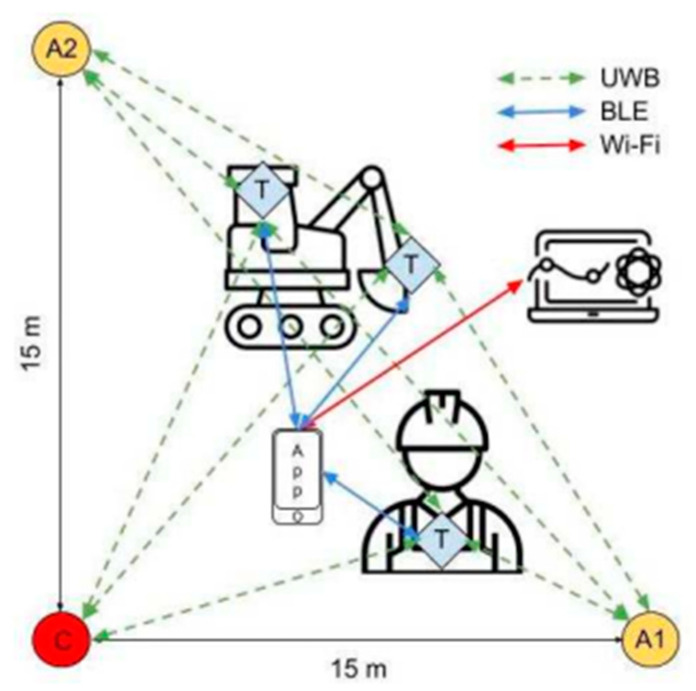
Infrastructure site configuration of the PWS with fixed anchors [20]. C: Coordinator, A1, A2: Anchors, T: Tags.

**Figure 7 sensors-23-09770-f007:**
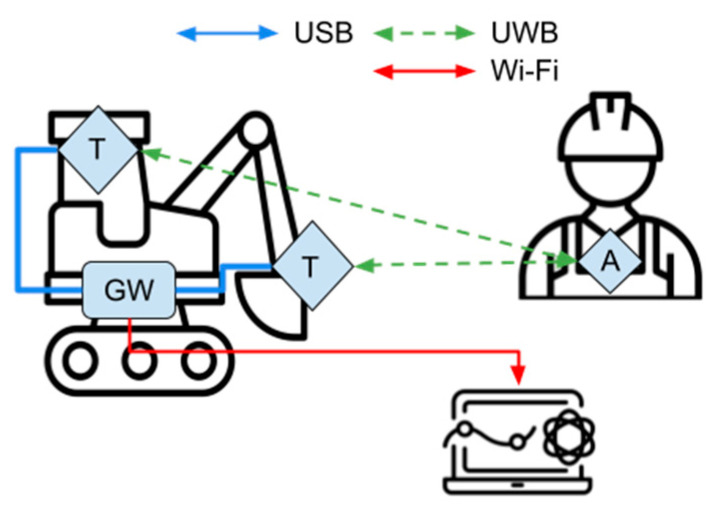
Infrastructure-less site configuration of the PWS without fixed anchors. A: Anchor, T: Tags, GW: Gateway.

**Table 1 sensors-23-09770-t001:** Technological solutions for proximity warning systems.

Type	Technological Solution	References (e.g.,)
Sensor-based locating	Global Positioning System (GPS)	[25]
	Radio Frequency Identification (RFID)	[26,27,28,29]
	GPS and Inertial Measurement Unit (IMU)	[19]
	Ultra-wideband (UWB)	[30,31]
	Bluetooth Low Energy RTLS	[36]
Vision-based sensing technologies		[32,33,34]

**Table 2 sensors-23-09770-t002:** Categories of personal data identified by the GDPR and filtered for the construction site use case and types of operations allowed.

Categories of Collected Personal Data	Types of Operations on Personal Data
Direct identification	A. Collecting
2. Indirect identification	B. Recording
3. Particular categories data	C. Organisation and Structuring
4. Health data	D. Conservation
5. Genetic data	E. Consultation
6. Biometric data	F. Use
7. Geolocalisation	G. Editing
	H. Extraction
	I. Fully Automated Decision Processing
	J. Profiling
	K. Pseudonymisation
	L. Anonymisation
	M. Communication by transmission
	N. Dissemination
	O. Comparison or Interconnection
	P. Limitation
	Q. Deletion
	R. Destruction
	S. Secure Copy (encrypted backup)
	T. Restore

**Table 3 sensors-23-09770-t003:** Data use case matched with the relevant categories of personal data and types of operations (note: x* means that data can be accessed by the data owner (i.e., the worker) and by the DPO).

1	2	3	4	5	6	7	Data Category	a	b	c	d	e	f	g	h	i	j	k	l	m	n	o	p	q	r	s	t
	x					x	Relative position of each actor	x	x		x	x*				x		x								x	
	x					x	Position of the vehicle	x	x		x	x*				x		x								x	
x						x	Definition of Area Location	x	x		x			x												x	x
x							Definition of Alert Entity	x	x		x			x												x	x
x							Definition of access restriction	x	x		x			x												x	x
	x					x	Access to Forbidden Area	x	x		x	x*				x		x								x	
	x					x	Transit in Forbidden Area	x	x		x	x*				x		x								x	
	x					x	Stop in Forbidden Area	x	x		x	x*				x		x								x	
x							Vehicle Digital Signature	x	x		x	x		x				x								x	x
	x					x	Vehicle Access Request	x	x		x	x*				x		x								x	

## Data Availability

Data are contained within the article.

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
