# Peer review of "Enhancing Safety on Construction Sites: A UWB-Based Proximity Warning System Ensuring GDPR Compliance to Prevent Collision Hazards"

_sensors, 2023, doi:10.3390/s23249770_

Round 1

Reviewer 1 Report

Comments and Suggestions for Authors

This manuscript describes the adoption of proximity warning systems (PWS) to reduce equipment collisions with material/workers/equipment. The paper presents the design of a prototype sensor-based commercial product for use at small to medium-sized construction sites.  Considerations included: cost, differences in construction locations, changing environments and data protection requirements. A low-cost, ultra-wideband (UWB) proximity detection system was selected and modeled using fixed anchors and without fixed anchors. In order to comply with EU General Data Protection Regulation (GDPR) the authors propose that a privacy-by-design approach would be adapted to all data collection such that person-identifiable data would not be collected and only anonymized or pseudoanonymized data would be used/ collected.  

1. Introduction: The introduction is a comprehensive review of the problem of equipment collisions/struck by incidents presenting a high risk for injury/death at construction sites. 

2. Research Background: This is a fairly comprehensive review of the need and types of real-time positioning systems in use or in development for construction site safety.  The authors should be sure and define all the different technology acronyms i.e. RTLS, GPS, RFID etc. when first mentioned. 

Sections 3 and 4. Research Methods and Materials and System Design and inplementation:  These sections are very detailed and describe proposed equipment used and  experimental approaches in great detail including two PWS approaches, one using fixed mapping of the construction site and locations/proximity of workers to equipment . The other approach described would  involve no fixed infrastructure and data collected would be strictly proximity data between workers and equipment. 

There are no actual results presented regarding either approach. No performance data of location accuracy, worker/equipment interaction data, no data on responses to proximity warning. 

The manuscript is an interesting review and thought exercise but really is more of a study design thought exercise.  It is unfortunate there are no actual results presented. 

Comments on the Quality of English Language

The quality of of English writing is very good overall.  There are a few instances of word-order, case inconsistencies that careful review by a native speaker would improve

Author Response

This manuscript describes the adoption of proximity warning systems (PWS) to reduce equipment collisions with material/workers/equipment. The paper presents the design of a prototype sensor-based commercial product for use at small to medium-sized construction sites.  Considerations included: cost, differences in construction locations, changing environments and data protection requirements. A low-cost, ultra-wideband (UWB) proximity detection system was selected and modeled using fixed anchors and without fixed anchors. In order to comply with EU General Data Protection Regulation (GDPR) the authors propose that a privacy-by-design approach would be adapted to all data collection such that person-identifiable data would not be collected and only anonymized or pseudoanonymized data would be used/ collected. 

  1. Introduction: The introduction is a comprehensive review of the problem of equipment collisions/struck by incidents presenting a high risk for injury/death at construction sites.

The Authors would link to thank the Reviewer for his/her positive comments about the state of the art analysis.

  1. Research Background: This is a fairly comprehensive review of the need and types of real-time positioning systems in use or in development for construction site safety. The authors should be sure and define all the different technology acronyms i.e. RTLS, GPS, RFID etc. when first mentioned.

The Authors revised the full paper to define the technology acronyms when first mentioned.

Sections 3 and 4. Research Methods and Materials and System Design and inplementation:  These sections are very detailed and describe proposed equipment used and  experimental approaches in great detail including two PWS approaches, one using fixed mapping of the construction site and locations/proximity of workers to equipment . The other approach described would  involve no fixed infrastructure and data collected would be strictly proximity data between workers and equipment.

The Authors would link to thank the Reviewer for his/her positive comments about the state of the art analysis.

There are no actual results presented regarding either approach. No performance data of location accuracy, worker/equipment interaction data, no data on responses to proximity warning.

The main target of the paper is to analyze and investigate privacy issues in the use of a technology capable of identifying in real time the position of workers in construction sites for security reasons. The authors proposed a privacy-by-design approach to address these privacy concerns. The evaluation of technology is beyond the scope of this paper. The Authors better clarify the scope of the paper in the introduction.

The ability of the proposed technology to accurately (and in real time) identify the position of workers in close proximity to construction machinery has already been investigated in a previous paper, and thus we prefer to refer to the paper [20].

  1. Bellagente, P. (2022). Assessment of UWB RTLS for Proximity Hazards Management in Construction Sites. In 2022 IEEE Sen-sors Applications Symposium (SAS), 1-6.

The manuscript is an interesting review and thought exercise but really is more of a study design thought exercise.  It is unfortunate there are no actual results presented.

As was noted in the response to the previous comment, the primary objective of the paper is to investigate and present a system architecture for the design of a proximity warning system that is GDPR EU compliant, rather than to investigate the performance of UWB technology. The feasibility of using UWB technology for location applications in construction sites has already been investigated in a previously paper, properly referred ([40]).

Comments on the Quality of English Language

The quality of of English writing is very good overall.  There are a few instances of word-order, case inconsistencies that careful review by a native speaker would improve

The Authors revised entirely the paper to fix language minor issues, as suggested by the Reviewer.

Reviewer 2 Report

Comments and Suggestions for Authors

The context of the study is explained in an accessible way. The case that the authors explore is quite interesting and of course important.

Missing reference for the international statistcocs on line 35.

Missing reference to previous studies on line 107.

The related work is a bit mixed up now. I would suggest the authors to have separate subsections about the existing studies and the existing technology. It would be usefull to see analysis of the state of the art based on the factors the authors consider for tehir work (see lines 89-91). A table here can also help better understand what has been done and compare it to your work.

Since this is a study utilising sensors and you have GDPR-compliant in. your title, I would expect literature review or at least discussion on that topic as well. See for example this work on GDPR-compliant sensor data sharing in smart cities:

Chhetri, T.R.; Kurteva, A.; DeLong, R.J.; Hilscher, R.; Korte, K.; Fensel, A. Data Protection by Design Tool for Automated GDPR Compliance Verification Based on Semantically Modeled Informed Consent. Sensors 2022, 22, 2763. https://doi.org/10.3390/s22072763

If you target the GDPR as main law then you need to elaborate more on the data protection ascpect in Section 2.1. Missing reference to the studies mention on line 209. 

Section 3 presents new infomation about an empirical study with end-users. Seems like here the focus was on understanding the willingness of people to share data and be monitored. Literature review on this is missing.

I am missing a clear step-by-step methodology. Now Section 3 reads more like a report. A diagram can help here as well.

If abbreviations were defined, the they should be used consistently later on in the text.

Missing reference to Decawave MDEK1001 Kit in Section 4.1.

Section 4.4 discusses privacy by design techniques but am overview of them is missing from the related work. 

Lines 469-473 present information that should be in the methdology. What is meant by digital representation on line 457?

The captions of Table 1 and Table 2 are separated from the tables. In Table 3, "Particular data categories" is generic and can refer to all the noted data in the table. Paraphrase to a more specific tem here.

Overall, good use case and idea. However, at its current state the paper is generic and lacks implementation details. Section 4 seems rushed. The paper needs restructuring of content and adding more analysis of related work and implementation details. Evaluation is missing. How does your work compare to the existing solutions? The limitations of your work are not clear.

Comments on the Quality of English Language

English language is understandable. Minor edits needed.

Round 2

Reviewer 1 Report

Comments and Suggestions for Authors

The authors were responsive to reviewer comments, The manuscript better defines that the intent of the paper is to discuss strategies to ensure that PWS need to consider that data collected should conform to the EU GDPR. The aim of the  authors to describe a privacy-by-design approach for the use of PWS is now more clear.